# A Directly Compressible Pregelatinised Sago Starch: A New Excipient in the Pharmaceutical Tablet Formulations

**DOI:** 10.3390/polym14153050

**Published:** 2022-07-28

**Authors:** Riyanto Teguh Widodo, Aziz Hassan, Kai Bin Liew, Long Chiau Ming

**Affiliations:** 1Department of Pharmaceutical Technology, Faculty of Pharmacy, Universiti Malaya, Kuala Lumpur 50603, Malaysia; 2Department of Chemistry, Faculty of Science, Universiti Malaya, Kuala Lumpur 50603, Malaysia; ahassan@um.edu.my; 3Department of Pharmaceutical Technology and Industry, Faculty of Pharmacy, University of Cyberjaya, Cyberjaya 63000, Malaysia; liewkaibin@cyberjaya.edu.my; 4PAP Rashidah Sa’adatul Bolkiah Institute of Health Sciences, Universiti Brunei Darussalam, Gadong BE1410, Brunei

**Keywords:** pregelatinised sago starch, excipient, tablet, direct compression

## Abstract

An excipient intended for direct compression in pharmaceutical tableting must show important features of flowability and compactibility. This study investigated pregelatinised sago starch as an excipient for direct compression tablets. Pregelatinised sago starch was prepared and characterised. Its powder bulk properties and performance in the tablet formulations with paracetamol as a model drug were compared against two commercial, directly compressible excipients, namely Avicel^®^ PH 101 and Spress^®^ B820. The results showed that pregelatinisation did not affect the chemical structure of sago starch, but its degree of crystallinity reduced, and X-ray diffraction pattern changed from C-type to A-type. Powder bulk properties of pregelatinised sago starch and Spress^®^ B820 were comparable, exhibiting better flowability but lower compactibility than Avicel^®^ PH 101. In the formulation of paracetamol tablets, pregelatinised sago starch and Spress^®^ B820 performed equally well, followed by Avicel^®^ PH 101 as indicated in Formulations 3, 2 and 1, respectively.

## 1. Introduction

Direct compression is a preferred method in the manufacturing of tablets as it offers simplicity and cost effectiveness, where tablets are formed by compression of the powder mixtures consisting of drug(s) and directly compressible excipients [1]. Demonstrating good flowability along with high compactibility is crucial for directly compressible tablet excipients [2].

Spray dried lactose, introduced in the early 1960s, was the first excipient designed for direct compression [2]. Since then, many excipients have been introduced to the pharmaceutical market as directly compressible excipients. For instance, corn starch has been successfully modified into a pregelatinised form and marketed as flowable and compressible starch with the commercial names Uni-pure^®^ DW and Uni-pure^®^ LD (National Starch & Chemical Co., Salisbury, NC, USA), Starch^®^ 1500 (Colorcon Inc., Harleysville, PA, USA) and Spress^®^ B820 (Grain Processing Corp., Muscatine, IA, USA). These aforementioned excipients are widely used as direct compression excipients in tablet formulations [3].

Sago starch is available abundantly in Malaysia with its main usage in food products [4]. In the pharmaceutical field, there has been extensive research into the utilisation of sago starch as a potential pharmaceutical excipient; however, research on the application of local sago starch, specifically as a directly compressible tablet excipient, has yet to be studied. Pregelatinised sago starch is sago starch that has been thermally modified into a pregelatinised form to be used as an excipient in direct compression tablets. This study aimed to prepare and characterise pregelatinised sago starch as a directly compressible tablet excipient. Its bulk powder properties and performances in tablet formulations containing paracetamol as a model drug were comparatively evaluated against two established directly compressible excipients, namely a product of pregelatinised corn starch and microcrystalline cellulose, commercially known as Spress^®^ B820 and Avicel^®^ PH 101, respectively.

## 2. Materials and Methods

### 2.1. Materials

A local sago starch (food grade, Nee Seng Ngeng & Sons Sago Industries Sdn Bhd, Sarawak, Malaysia), Spress^®^ B820 (Pre-gelatinised corn starch, GPC, Muscatine, IA, USA), Avicel^®^ PH101 (Microcrystalline cellulose, Fluka, Cork, Ireland), paracetamol BP powder and sodium starch glycolate (BP grade supplied by Euro Chemo-Pharma Sdn Bhd, Kuala Lumpur, Malaysia), magnesium stearate (Peter Greven Nederland C.V., Venlo, The Netherlands).

### 2.2. Preparation of Pregelatinised Sago Starch

Pregelatinised sago starch was prepared according to Widodo and Hassan, 2015 [5]. Aqueous slurry of 20% *W*/*V* sago starch was heated in a water bath (Grant SUB 36, Cheshire, England) at a predetermined pregelatinisation temperature of 65 °C with stirring at 700 RPM (WiseStir™ HD-30D, Daihan Scientific Co., Seoul, Korea) for 60 min. The resulting sago starch paste was dried in an Oven WTB Binder (Geprcifte Sicherheit, Tuttlingen, Germany) at 40 °C for 48 h. The dried mass was then powdered in a laboratory mill (MX-895M, National, Selangor, Malaysia). The dried starch powder was passed through a sieve (180 µm aperture) and stored in a tightly sealed white container prior to usage.

### 2.3. Spectroscopic Characterisations

#### 2.3.1. FT-IR Analysis

FT-IR spectra were obtained on an FT-IR spectrophotometer (Model: IFS66v/S) using a KBr disc containing 1% sample. The disc was scanned over a wave number range of 4000–400 cm^−1^ interval with a 4 cm^−1^ resolution.

#### 2.3.2. NMR Analysis

The NMR spectra were recorded at room temperature on a Broker AV400 solid-state NMR at a frequency of 100.6 MHz for ^13^C analyses. ^13^C NMR spectra were observed under cross polarisation magic angle sample spinning (CPMAS) with the spinning rate at 7 kHz. The chemical shift was determined using tetramethylsilane TMS (0 ppm) as an internal standard.

#### 2.3.3. X-ray Diffraction

The X-ray diffraction patterns were recorded using a D8 Advance X-ray Diffractometer (Bruker AXS, Karlsruhe, Germany) with CuK_α_ monochromatised radiation, running at 40 kV and 40 mA at ambient temperature. The scanning region of the diffraction angle 2θ was from 2° to 40° with step interval 0.02 and scanning rate of 2°/min. The total run time was 60 min. Degree of crystallinity of the starches was obtained by calculating the area of crystallinity in the X-ray diffractogram obtained [6].

### 2.4. Evaluation of Excipient Bulk Powders

#### 2.4.1. Particle Size Distribution, Shape and Surface Texture

Particle size distribution was determined using a light microscope (Nikon Eclipse 80i; Nikon Instruments Inc., Kanagawa, Japan). Mean projected diameter of 300 particles was calculated using NIS Element D2.30 computer software. The shape and surface texture of excipients were examined with a scanning electron microscope (SEM) (FEI Quanta 200 FESEM, Eindhoven, Holland) at an accelerating voltage of 10 to 12.5 kV.

#### 2.4.2. Moisture Content

Moisture content was determined according to The United States of Pharmacopeia [7]. Each of the excipients (1 g) was dried in the oven at temperature (100–105) °C for 4 h. The percentage loss in weight was calculated as moisture content. The results were reported in triplicate.

#### 2.4.3. Bulk Density (*ρ*_0_) and Tapped Density (*ρ*_t_)

The bulk density (*ρ*_0_) and tapped density (*ρ*_t_) of the excipients were determined by a tapped density tester (JV2000; Dr.Schleuninger Pharmaton AG, Solothurn, Germany). A 250 mL glass cylinder was filled with 100 g of powder sample and placed on top of the tapped density tester, and bulk volume was recorded. The cylinder was then tapped 1000 times to a constant volume and the tap volume was recorded. Bulk and tapped densities were calculated based on the ratio of weight to volume. The procedure was done in triplicate.

#### 2.4.4. Flow Properties

The excipients’ flow properties were evaluated by measuring their angle of repose through fixed height cone method. A quantity of sample powder (100 mg) was filled into the steel funnel (BEP-Auto, Dr. Schleuniger Pharmaton, Nottingham, UK) and allowed to completely discharge from the funnel. The height (Y) and diameter (X) of the pile of powder that formed were measured. The angle of repose α (°) was calculated according to Formula (1). The results were reported in triplicate.
Angle of repose, tan α = 2Y/X(1)

#### 2.4.5. Compactibility

To evaluate the excipients compactibility, round flat-faced compacts were individually prepared by compressing each of the excipients in an Enerpac GA3 Single Punch Machine (MTCM-1, Globepharma Inc., New Brunswick, NJ, USA) at ten different compression pressures (from 20 to 200 MPa) and repeated three times for each level of compression pressure. The compacts were weighed and measured for diameter, thickness and hardness (Model 6D, Dr. Schleuniger Pharmatron, Manchester, NH, USA) after storage for 24 h. The tensile strength *T* of the compact was then calculated using Formula (2).
(2)T=2F/πdt
where *F*, *d* and *t* are the hardness, diameter and thickness of the compact, respectively.

#### 2.4.6. Loading Capacity

Paracetamol is poorly compressible; thus, it is suitable to be used as a model drug for loading capacity determination [1,8]. Paracetamol powder was added to the excipient at concentrations of 0%, 10%, 20%, 30%, 40%, 50%, 60% and 70% and mixed in a cube mixer (Type KB 15/UG, Hensenstamm, Germany) attached to a motor drive (Type AR401, ERWEKA, Hensenstamm, Germany) for 5 min at 200 rpm. The mixture was compressed into compacts (300 mg) using a single punch compaction machine (MTCM-1, Globepharma Inc., New Brunswick, NJ, USA) and die/punch set of 8.33 mm size at different compression pressures, ranging from 20 to 120 MPa. The process was repeated three times for each level of compression pressure. Compacts were stored for 24 h post compression. The compact’s hardness and dimension (diameter and thickness) was measured, and the compact’s tensile strength was calculated. The loading capacity was then determined [9,10].

#### 2.4.7. Development of Model Drug Formulation

Formulations of tablets containing paracetamol as a model drug were developed as shown in Table 1. All ingredients (except magnesium stearate) were mixed with paracetamol for 10 min in a cube mixer at 200 rpm. Magnesium stearate was then added and mixed for another 3 min. The powder mixtures were directly compressed into tablets using a single punch compaction machine (MTCM-1, Globepharma Inc., New Brunswick, NJ, USA), with the die/punch set to 10.00 mm diameter at a predetermined compression pressure to obtain tablets with a similar degree of hardness at about 100 N. The tablets produced were stored in a desiccator for 24 h prior to further evaluation of uniformity of weight, hardness, dimension, friability, disintegration and dissolution [7,11,12].

#### 2.4.8. Short-Term Accelerated Stability Study

Short-term accelerated stability study was conducted according to International Conference on Harmonization [13]. Tablets produced from the selected formulations were kept in closed glass containers at 40 ± 2 °C/75 ± 5% RH for 3 months and 6 months, and the tablets were evaluated in the same manner as above for uniformity of weight, hardness, dimension, friability, disintegration and dissolution.

#### 2.4.9. Statistical Analysis

The differences in means of data were analysed using *t*-test and analysis of variance (ANOVA) tests. Results with *p* < 0.05 were considered to be statistically significant.

## 3. Results and Discussion

### 3.1. Spectroscopic Characterisation

#### 3.1.1. FT-IR Analysis

FT-IR spectra were used to verify the changes in chemical structure as a result of pregelatinisation. The FT-IR spectra analyses revealed that there were no new sharp bands found in the pregelatinised sago starch (Figure 1), indicating that pregelatinisation did not change the chemical structure of sago starch.

The characteristics of strong, broad bands between 927 cm^−1^ and 1200 cm^−1^ with three peaks at 980 cm^−1^, 1084 cm^−1^ and 1166 cm^−1^, associated with (C-O-H), (C-C) and (C-O), represent a glycosidic linkage. Another characteristic of sharp bands are the three peaks at 1649 cm^−1^, 2920 cm^−1^ (refer to the C-H stretching of starch molecules) and 3422 cm^−1^ (refer to O-H group of starch). The two sharp bands at 1645 cm^−1^ and 1411 cm^−1^ are characteristics of bound water and C-H bonding present in the starch molecules [14,15].

#### 3.1.2. C-NMR Analysis

From the ^13^C-NMR spectra (Figure 2), it was shown that there are no significant chemical structure changes for sago starch after pregelatinisation. Chemical shift (δ) variations shown from the studies were of minor significance. This could be due to when the starch is heated in excess water, the crystalline structure of starch granule is disrupted [16,17].

In the presence of water with heat, the water molecule tends to diffuse and be taken up by the starch granules. This resulted in the uncoiling and dissociation of the double helical structure of linear starch chains, leading to the loss of the organised structure of starch. However, the basic monomer structure of the D-glucose unit found in both amylose and amylopectin components of starch remained uninterrupted. Thus, the ^13^C-NMR spectra of pregelatinised sago starch showed no chemically important differences in comparison with sago starch spectra. This suggests that pregelatinisation of starch will only disrupt the hydrogen bonding between linear starch chains, leading to disorganisation of the double helical structure of starch, but the basic monomer backbone will be unaffected [1].

#### 3.1.3. X-ray Diffraction

X-ray diffraction patterns of sago starch and pregelatinised sago starch are presented in Figure 3. It was observed that sago starch exhibits characteristics of a C-type diffraction pattern, characterised by a weak peak at 2θ: 5.60° and strong peaks at 2θ: 15°, 17°, 18° and 23° [18,19], while pregelatinised sago starch showed an A-type pattern, characterized by strong peaks at 2θ: 15°, 17°, 18° and 23° [20].

The peaks of sago starch appeared sharper compared to its pregelatinised forms, indicating that sago starch has a larger crystalline region. This study showed that pregelatinisation significantly reduced the degree of crystallinity (*p* < 0.05) from (52.00 ± 1.03) to (35.11 ± 1.10)%.

### 3.2. Evaluation of Excipient Bulk Powders

#### 3.2.1. Particle Analysis, Moisture Content, Bulk and Tap Densities and Flow Properties

Results of the excipient bulk powder evaluations are presented in Figure 4 and Table 2. Avicel^®^ PH 101 exhibits a fibrous rodlike structure and a far from spherical shape, while Spress^®^ B820 and pregelatinised sago starch appear as granules with irregular shape and rough surface structure. Their mean diameters were (56.70 ± 11.51), (89.30 ± 20.29) and (88.00 ± 18.98) µm for the moisture content at (5.19 ± 0.06), (9.91 ± 0.02) and (10.39 ± 0.41) %, respectively.

Bulk density of Spress^®^ B820 was recorded at (0.64 ± 0.00) g/cm^3^, pregelatinised sago starch at (0.52 ± 0.01) g/cm^3^ and Avicel PH 101 at (0.35 ± 0.00) g/cm^3^. The highest bulk density of Spress^®^ B820 was due to its small particles that lie in between the larger ones to form lower bulk volume, while the lowest bulk density of Avicel PH 101 resulted from its fibrous rodlike structure, where particles of Avicel PH 101 were packed in an irregular manner, thus demonstrating large voids among the particles. The mean value of tap density obtained for Spress^®^ B820 was (0.71 ± 0.00) g/cm^3^, followed by pregelatinised sago starch of (0.61 ± 0.01) g/cm^3^ and Avicel PH 101 of (0.44 ± 0.00) g/cm^3^. Tapping actions allowed smaller particles to slip between the larger ones producing a lower bulk volume. Consequently, tap density is always higher than bulk density, and thus the order could be similar to the bulk density. Spress^®^ B820 and pregelatinised sago starch showed angles of repose of (30.23 ± 0.46) and (30.37 ± 0.23)°, respectively, and were rated as having excellent flow, and Avicel PH 101, with an angle of repose of (41.87 ± 0.51)°, was rated as passable [21]. This study found that the flowability of pregelatinised sago starch and Spress^®^ B820 was excellent and significantly influenced by their particle size and moisture content with very strong correlation (*p* < 0.05, R^2^ = 0.996). The poor flow properties of Avicel^®^ PH 101 resulted from particle size (*p* < 0.05) and its rodlike structure, where entanglements between particles arise and hamper the flowability.

#### 3.2.2. Compactibility and Loading Capacity

Compaction profile of Figure 5 shows strong linear relationships, indicating that these three excipients undergo plastic deformation [3], with R^2^ values at 0.8884, 0.9874 and 0.9961 for Avicel PH 101, Spress^®^ B820 and pregelatinised sago starch, respectively. Avicel PH 101 compacts were found to have the highest radial tensile strength at any of the compression pressures, followed by Spress^®^ B820 and pregelatinised sago starch. This could be due to the fast formation of interparticle bonding under compression and the presence of optimum moisture content (5.19%) within the porous structure of Avicel^®^ PH 101, which acts as an internal lubricant to smooth the flow within the individual microcrystalline particles during plastic deformation. Hence, Avicel PH 101 demonstrated the greatest degree of plastic deformation and the highest compactibility among the excipients.

Figure 5 illustrates the compaction profiles of Avicel PH 101, Spress^®^ B820 and pregelatinised sago starch upon paracetamol loading at various concentrations, where the tensile strength of the compacts reduced significantly (*p* < 0.05) with increasing paracetamol concentration. Consequently, the area under the plotted curves (AUC) of the tensile strength versus compression pressure decreased with increasing paracetamol content.

To determine loading capacity of the excipient, plots of the area ratio versus percentage (%) of paracetamol were constructed (Figure 6). Linear regression and back extrapolation to zero area ratios of the plots gave the loading capacity values [10], which reflect the minimum amount of excipient required to form a tablet with the poorly compressible active ingredient, and thus the higher value is preferable. The plots show loading capacity of Avicel PH 101, Spress^®^ B820 and pregelatinised sago starch relative to paracetamol reached up to 70.16%, 59.16% and 60.97%, respectively.

### 3.3. In Vitro Evaluations of Developed Formulation of Paracetamol Tablets

#### 3.3.1. Uniformity of Weight, Friability and Dimension

All of the formulations produced tablets in compliance with The United States of Pharmacopeia [7] for uniformity of weight and friability. Diameter of the tablets was similar according to the die and size of punches used in this study. Thickness of the tablets were significantly different (*p* < 0.05) for each of the formulations due to the variation in compression pressure applied to produce tablets. The results are summarised in Table 3.

#### 3.3.2. Hardness

Paracetamol tablets in this study were designed to have an ideal hardness between 90 and 110 N, with the intention to minimise the effects resulting from various degrees of hardness of the tablets on their disintegration and dissolution properties [11]. It was found that paracetamol tablets formulated with Avicel PH 101 (Formulation 1), Spress^®^ B820 (Formulation 2) and pregelatinised sago starch (Formulation 3) needed compression pressures of 40, 160 and 120 MPa, respectively, to produce tablets with the hardnesses of 90 to 110 N. Formulation 1 required the lowest compression pressure due to the inclusion of Avicel PH 101. Although Spress^®^ B820 exhibited higher plasticity than pregelatinised sago starch, Formulation 2 required higher compression pressures than Formulation 3, which could be related to the higher magnesium stearate sensitivity and larger average particle size of Spress^®^ B820 than pregelatinised sago starch. Excipients that were sensitive to magnesium stearate would soften the tablets [2], while excipients with large particle size only provided a small surface area to form bonding between the particles [3]. When Avicel PH 101 was introduced, as shown in Formulations 4 and 5 (Table 2), Formulation 4 needed a lower compression pressure of 140 MPa than Formulation 2, while Formulation 5 required the same compression pressure as Formulation 3, but it produced paracetamol tablets with higher hardness.

#### 3.3.3. Disintegration

All the tablets disintegrated in less than 2 min (Table 3) (*p* < 0.05), thus meeting The United States of Pharmacopeia specifications for uncoated tablets. This shows that the three directly compressible excipients possessed good disintegration properties. The rapid disintegration of Formulation 1 was due to the ability of water to penetrate into the porous hydrophilic tablet matrix of Avicel^®^ PH 101, subsequently disrupting the hydrogen bonding of the particles, and hence weakening the strength of the tablets [22]. Penetration of water into the Avicel^®^ PH 101 matrix was much faster than into Spress^®^ B820 or pregelatinised sago starch matrices of Formulation 2 or 3, respectively. Formulations 2 and 4 exhibited longer disintegration time than Formulations 3 and 5, respectively. It is possible that Spress^®^ B820 was more sensitive to the magnesium stearate lubricant than pregelatinised sago starch; as such, the resulting paracetamol tablets were more hydrophobic and less accessible by the aqueous medium [23]. The shorter disintegration time of Formulations 4 and 5 than Formulations 2 and 3, respectively, was due to the inclusion of Avicel^®^ PH 101 and sodium starch glycolate in the formulations as complimentary disintegration agents. The porous structure of Avicel^®^ PH 101 accelerated the water penetration into the tablet matrix and hastened the swelling of Spress^®^ B820, pregelatinised sago starch and sodium starch glycolate to disintegrate the tablets [24].

#### 3.3.4. Dissolution

Results of the dissolution tests and their efficiency are presented in Figure 7 and Table 3. Formulation 1 did not release 80% of paracetamol within 30 min as required by the The United States of Pharmacopeia for uncoated tablets [7]. This was due to the formation of a viscous diffusion layer by Avicel^®^ PH 101, making paracetamol slow to diffuse into the bulk of the dissolution medium [23]. Formulations 2 and 3 released more than 80% of paracetamol within 20 min and released 100% of the paracetamol within 25 min. Formulation 4 and Formulation 5 showed excellent release of paracetamol as a result of including super disintegrant sodium starch glycolate (*p* < 0.05) in the formulations. Both formulations released more than 80% of paracetamol within 5 min, and within 15 min they released 100% and 99.95% of paracetamol, respectively. The area under dissolution curves found that Formulation 1 showed the lowest dissolution efficiency (40.14%), followed by Formulation 2 (69.50%), Formulation 3 (71.25%), Formulation 5 (86.36%) and Formulation 4 (86.55%).

A difference in dissolution efficiency below 10% means dissolution profiles are equivalent and could be similar in bio-equivalency [11]. Hence, paracetamol tablets of Formulation 1 are dissimilar in dissolution profile and bio-equivalency than those of Formulation 2 and Formulation 3. In comparison, Formulation 2 versus Formulation 3 and Formulation 4 versus Formulation 5 showed only 1.75 and 0.19% differences in dissolution efficiency, respectively. This indicated that paracetamol tablets formulated with the same concentration of Spress^®^ B820 and pregelatinised sago starch produced tablets with similar dissolution profiles and may also be similar in bio-equivalency.

#### 3.3.5. Short-Term Accelerated Stability Study

Formulations 4 and 5 exhibited a superior release profile among all of the formulations; therefore, they were selected for further accelerated stability study, with the tablets produced on day 1 used as references. The results showed that all evaluated parameters such as uniformity of weight, dimension, hardness, friability, disintegration and dissolution for Formulations 4 and 5 after 3 and 6 months of storage at the accelerated stability study conditions were found to be similar (*p* < 0.05) to those of the references, respectively (Table 2 and Figure 7). This indicated that the formulations were compatible and stable.

## 4. Conclusions

This study indicates that pregelatinised sago starch has the potential to be a new alternative for a directly compressible excipient. Its flow properties were excellent and performed well in the tablet formulations using paracetamol as a model of active ingredient, with the resulting tablets exhibiting good stability, satisfactory disintegration time and dissolution properties.

## Figures and Tables

**Figure 1 polymers-14-03050-f001:**
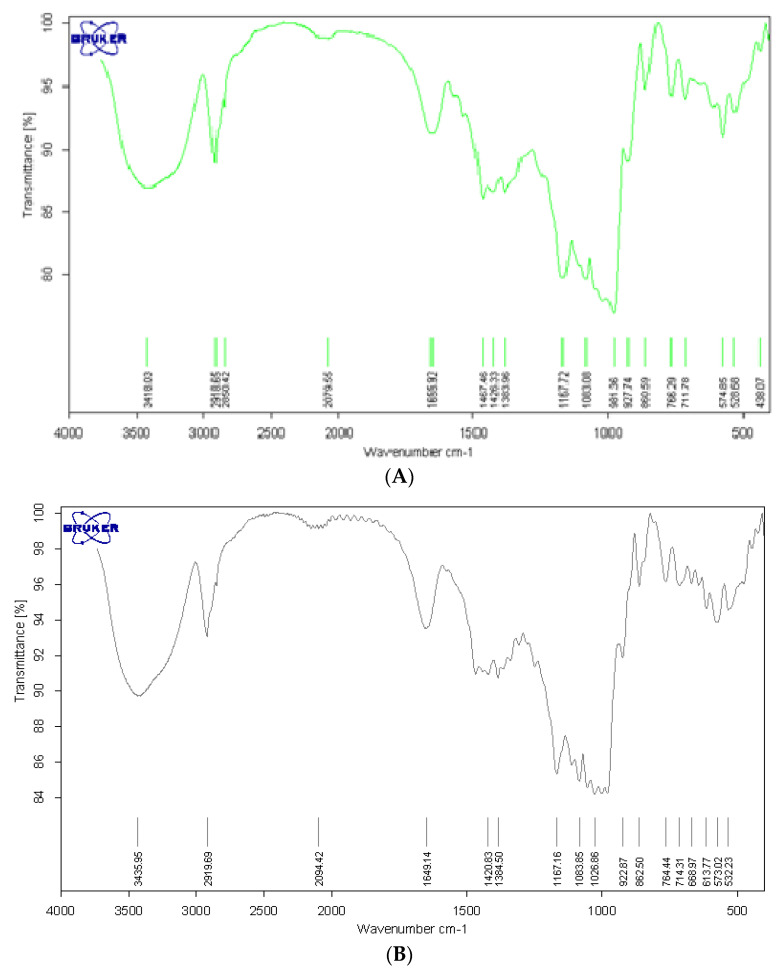
FTIR spectra of (**A**) sago starch; (**B**) pregelatinised sago starch.

**Figure 2 polymers-14-03050-f002:**
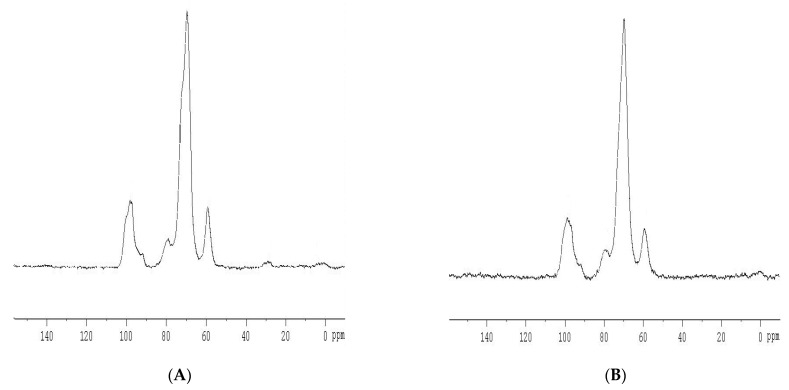
^13^C-NMR Spectra of (**A**) sago starch (δ: 98.0, 78.9, 69.7 and 59.2 ppm); (**B**) pregelatinised sago starch (δ: 98.6, 79.1, 69.7 and 59.4 ppm).

**Figure 3 polymers-14-03050-f003:**
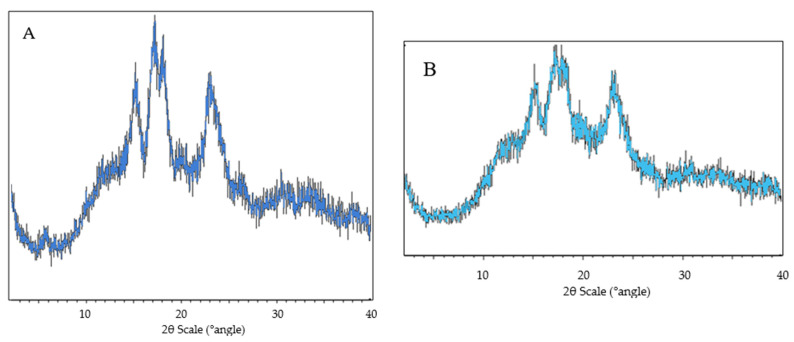
X-ray diffractogram of (**A**) sago starch; (**B**) pregelatinised sago starch.

**Figure 4 polymers-14-03050-f004:**
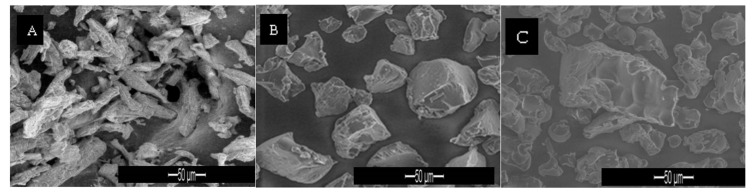
SEM images of (**A**) Avicel^®^ PH 101; (**B**) Spress^®^ B820; (**C**) pregelatinised sago starch.

**Figure 5 polymers-14-03050-f005:**
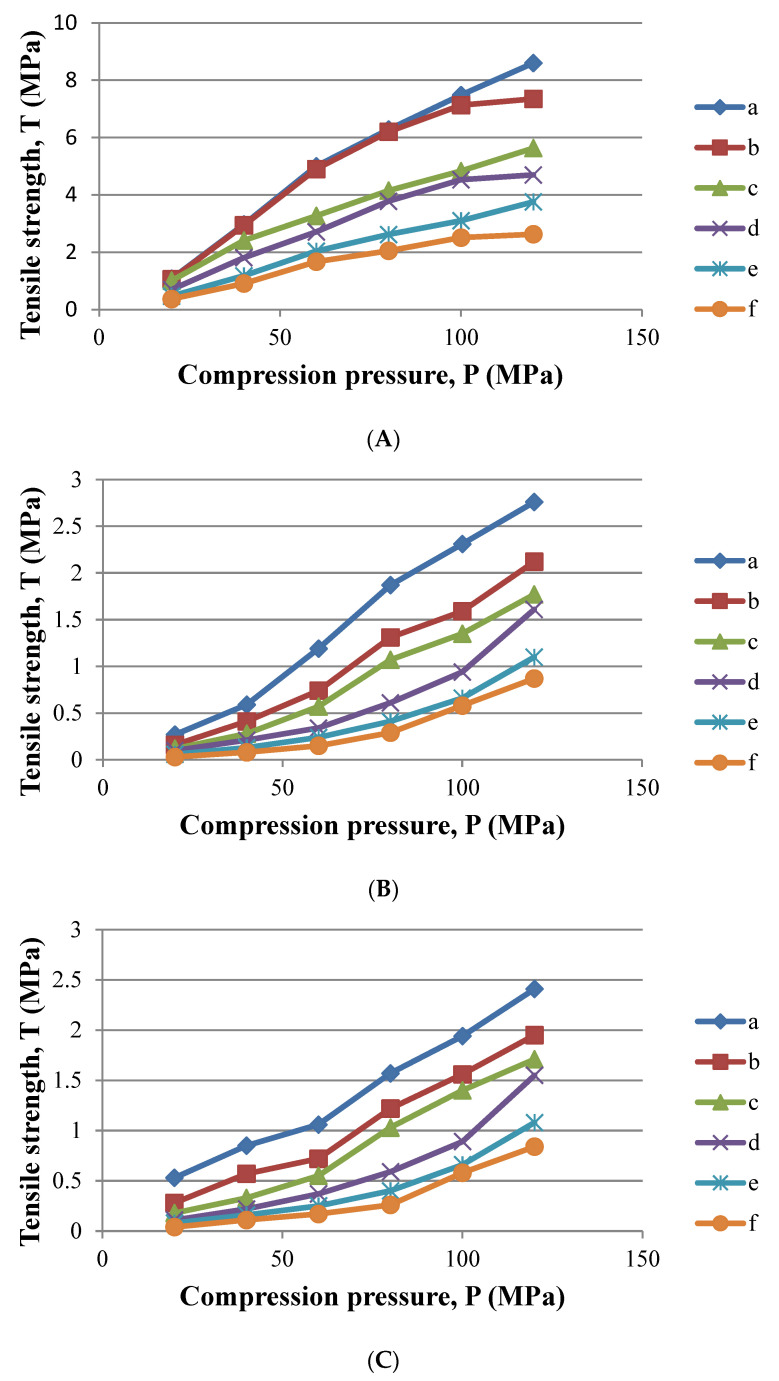
Compaction profile of the excipients (**A**) Avicel PH 101; (**B**) Spress^®^ B820; (**C**) pregelatinised sago starch and paracetamol mixture of (a) 0% *w*/*w* paracetamol; (b) 10% *w*/*w* paracetamol; (c) 20% *w*/*w* paracetamol; (d) 30% *w*/*w* paracetamol; (e) 40% *w*/*w* paracetamol; (f) 50% *w*/*w* paracetamol.

**Figure 6 polymers-14-03050-f006:**
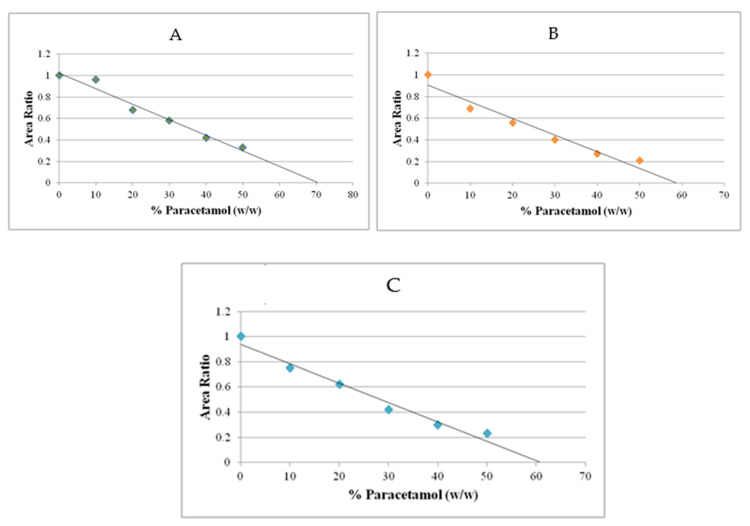
Plot area ratio of (**A**) Avicel PH 101 and concentration of paracetamol; (**B**) Spress^®^ B820 and concentration of paracetamol; (**C**) pregelatinised sago starch versus concentration of paracetamol.

**Figure 7 polymers-14-03050-f007:**
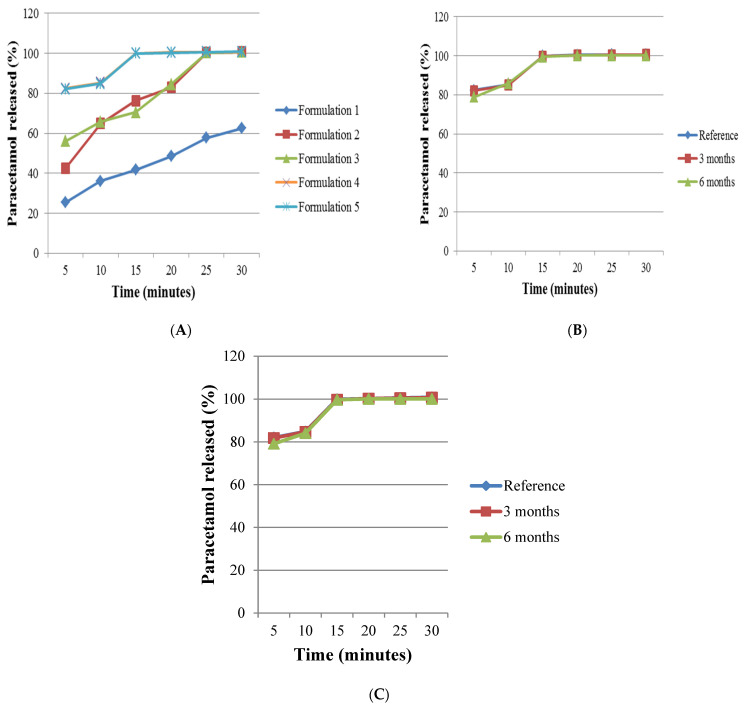
(**A**) Plot of percentage of paracetamol released over time for the dissolution test of each formulation; (**B**) Comparison of paracetamol released (%) between the reference, 3 months and 6 months of storage at conditions 40 ± 2 °C/75 ± 5% RH for Formulation 4; (**C**) Comparison of paracetamol released (%) between the reference, 3 months and 6 months of storage at conditions 40 ± 2 °C/75 ± 5% RH for Formulation 5. Note: *n* = 6 for all formulations.

**Table 1 polymers-14-03050-t001:** Formulation of paracetamol tablets.

Ingredient	Formulation								
	1	2	3	4			5		
				D1	3 M	6 M	D1	3 M	6 M
Paracetamol	120.0	120.0	120.0	120.0	120.0	120.0	120.0	120.0	120.0
Avicel PH 101	229.1	-	-	70.0	70.0	70.0	70.0	70.0	70.0
Spress^®^ B820	-	229.1	-	145.1	145.1	145.1	-	-	-
Pregelatinised sago starch	-	-	229.1	-	-	-	145.1	145.1	145.1
Sodium starch glycolate	-	-	-	14.0	14.0	14.0	14.0	14.0	14.0
Magnesium stearate	0.9	0.9	0.9	0.9	0.9	0.9	0.9	0.9	0.9

**Table 2 polymers-14-03050-t002:** Mean particle diameter, moisture content, densities, angle of repose and AUC _T vs P_ of each excipient bulk powder.

Excipients	Avicel PH 101	Spress^®^ B820	Pregelatinised Sago Starch
Mean particle diameter (µm) ± SD ^a^	56.70 ± 11.51	89.30 ± 20.29	88.00 ± 18.98
Moisture content (%) ± SD ^b^	5.19 ± 0.06	9.91 ± 0.02	10.39 ± 0.41
Bulk density (g/cm^3^) ± SD ^b^	0.35 ± 0.00	0.64 ± 0.00	0.52 ± 0.01
Tap density (g/cm^3^) ± SD ^b^	0.44 ± 0.00	0.71 ± 0.00	0.61 ± 0.01
Angle of repose, α (°) ± SD ^b^	41.87 ± 0.51	30.23 ± 0.46	30.37 ± 0.23
AUC _T vs P_	1270.6	446.8	392.7

Note: ^a^ *n* = 300 for each excipient, ^b^ *n* = 3 for each excipient, AUC _T vs P_: area under curve of tensile strength versus compression pressure.

**Table 3 polymers-14-03050-t003:** Evaluation of paracetamol tablets and stability study results of Formulations 4 and 5 after 3 months and 6 months of storage at conditions 40 ± 2 °C/75 ± 5% RH.

Evaluation	Formulation								
	1	2	3	4			5		
				D1	3 M	6 M	D1	3 M	6 M
Thickness (mm) ^a^	3.96 ± 0.05	3.55 ± 0.03	3.66 ± 0.03	3.59 ± 0.02	3.60 ± 0.01	3.61 ± 0.02	3.62 ± 0.01	3.62 ± 0.02	3.62 ± 0.03
Diameter (mm) ^a^	9.90 ± 0.01	9.88 ± 0.01	9.90 ± 0.01	9.88 ± 0.00	9.90 ± 0.02	9.90 ± 0.02	9.90 ± 0.00	9.91 ± 0.02	9.92 ± 0.03
Hardness (N) ^a^	92.91 ± 4.37	103.506. ± 72	103.50 ± 7.46	98.70 ± 5.17	100.00 ± 2.21	100.40 ± 3.60	109.10 ± 5.00	107.40 ± 2.55	106.50 ± 2.37
Uniformity of weight (mg) ^b^	347.82 ± 1.00	348.79 ± 0.76	350.05 ± 1.29	350.06 ± 0.93	348.79 ± 0.66	348.68 ± 0.89	349.40 ± 1.05	348.97 ± 0.76	348.75 ± 0.55
Friability (%)	0.38	0.48	0.51	0.54	0.54	0.58	0.43	0.46	0.46
Disintegration time (min) ^c^	0.23 ± 0.05	1.41 ± 0.16	1.33 ± 0.12	0.57 ± 0.37	0.56 ± 0.04	0.58 ± 0.06	0.44 ± 0.18	0.47 ± 0.03	0.45 ± 0.04
Dissolution efficiency (%)	40.14	69.50	71.25	86.34	87.76	85.75	86.36	86.23	85.53

Note: ^a^ *n* = 10 for both formulations, ^b^ *n* = 20 for both formulations and ^c^ *n* = 6 for both formulations, D1 = tablets produced at day-1 used as a reference, 3 M = tablets after 3 months storage, 6 M = tablets after 6 months storage.

## Data Availability

Not applicable.

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
