# Peer review of "A Directly Compressible Pregelatinised Sago Starch: A New Excipient in the Pharmaceutical Tablet Formulations"

_polymers, 2022, doi:10.3390/polym14153050_

Round 1
Reviewer 1 Report
The overall manuscript is well written however the manuscript is heavily loaded with information, including the key objectives from the authors previous research article "Widodo, Riyanto Teguh, and Aziz Hassan. "Compression and mechanical properties of directly compressible pregelatinized sago starches." Powder Technology 269 (2015): 15-21."
There are some additional analytical techniques mentioned, but there is a high percentage of overlap, including implementation of similar data and SEM images, without proper citation. The authors are requested to trim down the previously published section and focus primarily on the new findings in this paper.
Suggestion would be to trim the article to a form with particular focus on
Reviewer 2 Report
Paper is interesting but more detailed research work must be included
-uniformity of weight, hardness, dimension, friability, disintegration and dissolution test must be explained in material and methods
- dissolution assays must be performed in the three standard media pH1,2; ph 4,5 and pH 6,8 or, at least in a simulated digestion process during 2-2,5 hours
- in order to evaluate the stability profile a time 0 months, 3moths and 6 months of all formulations must be included
- conclusion must be more detailed
- Quality of figures can be improved specially figure 5c in which c appears superimposed on the legend of the y-axis
Round 2
Reviewer 1 Report
please ensure all work from previous publications be clearly cited, including individual image files
Reviewer 2 Report
Authors have followed the suggestions and paper has improved. I think the new version is suitable por publications